# Comparison of Trunk Motion between Moderate AIS and Healthy Children

**DOI:** 10.3390/children9050738

**Published:** 2022-05-18

**Authors:** Lucas Struber, Vincent Nougier, Jacques Griffet, Olivier Daniel, Alexandre Moreau-Gaudry, Philippe Cinquin, Aurélien Courvoisier

**Affiliations:** 1TIMC, University Grenoble Alpes, CNRS, UMR 5525, VetAgro Sup, Grenoble INP, CHU Grenoble Alpes, 38000 Grenoble, France; lucas.struber@univ-grenoble-alpes.fr (L.S.); vincent.nougier@univ-grenoble-alpes.fr (V.N.); odaniel@chu-grenoble.fr (O.D.); alexandre.moreau-gaudry@univ-grenoble-alpes.fr (A.M.-G.); pcinquin@chu-grenoble.fr (P.C.); 2Grenoble Alps Scoliosis and Spine Center, Grenoble Alps University Hospital, Bvd de la Chantourne, CEDEX 09, 38043 Grenoble, France; jgriffet@chu-grenoble.fr

**Keywords:** idiopathic scoliosis, motion analysis, kinematics, posture, spine

## Abstract

Analysis of kinematic and postural data of adolescent idiopathic scoliosis (AIS) patients seems relevant for a better understanding of biomechanical aspects involved in AIS and its etiopathogenesis. The present project aimed at investigating kinematic differences and asymmetries in early AIS in a static task and in uniplanar trunk movements (rotations, lateral bending, and forward bending). Trunk kinematics and posture were assessed using a 3D motion analysis system and a force plate. A total of fifteen healthy girls, fifteen AIS girls with a left lumbar main curve, and seventeen AIS girls with a right thoracic main curve were compared. Statistical analyses were performed to investigate presumed differences between the three groups. This study showed kinematic and postural differences between mild AIS patients and controls such as static imbalance, a reduced range of motion in the frontal plane, and a different kinematic strategy in lateral bending. These differences mainly occurred in the same direction, whatever the type of scoliosis, and suggested that AIS patients behave similarly from a dynamic point of view.

## 1. Introduction

Adolescent idiopathic scoliosis (AIS) is a structural, lateral, rotated curvature of the spine that affects 1–3% of children and arises around puberty [1]. AIS is a 3D spine deformity that has been well described from static standardized standing position X-rays that do not reflect the daily life of our patients [2,3]. Nowadays, motion capture analysis is used in routine practice to better describe gait and posture in several pathologies in the field of pediatric orthopedics. Trunk motion analysis is less common but would be relevant for a better understanding of AIS biomechanics. 

Recent studies have exhibited the variation of trunk motion according to both the severity [4] and type of scoliosis [5]. AIS has also been associated with postural instability [6,7], sensory integration deficits [7,8], and alteration of gait pattern combined with a reduced range of motion in the upper body and lower extremities [9,10]. However, observed differences are not consistent, relatively small in comparison to a healthy population, and most of these studies were limited to one type of curve or mixed up all curve types and Cobb angles. 

A better methodology of trunk motion analysis and patient selection seems important in order to come up with a more complete description of AIS. The selection of moderate curve magnitude would also help the detection of singularities that are already present at an early stage of the deformity. 

The main objective of this study was to compare trunk motion and posture between mild AIS and healthy children when performing simple uniplanar upper body movements. 

## 2. Materials and Methods

### 2.1. Participants

In total, 47 adolescents aged between 9 and 16 years voluntarily participated in the study (Table 1). The study took place in Grenoble Alps Scoliosis and Spine Center (Grenoble Alps University Hospital, Grenoble, France). A total of 15 healthy girls composed the control group (HS) were recruited from local schools, 15 AIS girls with left lumbar main curve composed the AIS-LL group, and 17 AIS girls with right thoracic main curve composed the AIS-RT group, respectively, type 1 and 5 according to Lenke classification [11]. For the two AIS groups, radiological data, Cobb angle, and Risser sign [12] were evaluated from standard standing anteroposterior radiography. In the AIS groups, the patients included in the study were not braced or were included in the study before bracing. The local ethics committee (Comité de Protection des Personnes Sud-Est V) approved this research (Ref. CPP 14-CHUG-14), and all methods were performed in accordance with the Declaration of Helsinki [13] and written informed consent was obtained from all the subjects and their parents.

### 2.2. Experimental Tasks

The experimental study was performed in the Motion Capture Lab located in our hospital. The lab is equipped with a 3D motion analysis system (Codamotion^®^ Rothley, UK) including 4 infrared cameras and a force plate (Accugait, AMTI^®^ Watertown, MA, USA). Participants started the experiment by performing a static task. They stood upright and barefoot, with their arms along the body, and their heels 5 cm apart with the internal edges of the feet making a 30° angle. They were instructed to focus on a cross target located at eye level and to remain in a stable relaxed posture for 30 s. A total of 4 trials were performed. Then, starting upright, participants performed three dynamic tasks involving the upper body: (1) forward maximal bending with arms along the body, (2) right and left maximal lateral bending with arms along the body, and (3) right and left maximal axial rotations with arms crossed on the chest. Each task was repeated 4 times.

### 2.3. Data Recording

A force plate recorded center of feet pressure (CoP) displacements (Accugait, AMTI^®^). Participants’ upper body motion was assessed using a 3D motion analysis system (Codamotion^®^) including 4 infrared cameras. Each participant was equipped with 13 reflective markers fixed on specific anatomical landmarks. The detailed placement of markers and test processing are described in Figure 1. The same trained operator (orthopedic surgeon) positioned all markers for all patients and used the same following method to choose the markers’ position. T1 and L5 were easily located through simple palpation. T12-L1 (thoraco-lumbar junction) and the apex of the kyphosis and lordosis were determined by counting the vertebrae starting from T1. The recording systems were synchronized and sampled at 100 Hz. All data were low-pass filtered with a zero-lag fourth-order Butterworth filter with a 10 Hz cut-off frequency.

### 2.4. Data Analysis

Computation of all parameters and statistical analyses were performed with Matlab^®^ software (Montbonnot-Saint-Martin, France). In the static condition, the participants’ stability was evaluated through a 95% confidence ellipse area, the mean speed of CoP, and sample entropy assessing the amount of attention invested in postural control [14]. Complying with the optimization criterion, we chose m = 3 and r = 0.04 in the entropy algorithm. The pelvis and thoracic rigid bodies orientations, defined by Euler angles following the International Society of Biomechanics (ISB) recommendations [15]—with respect to the reference frame—were investigated. To extend this evaluation and assess relevant spine morphology, different spine and acromial line angles were also computed [16] (see Figure 1 and legend for precise definitions). Note that the anatomical landmarks on the spine were chosen relatively to normal kyphosis and lordosis and not to scoliotic curvature, so as to enable direct comparison between HS and AIS. Regarding dynamic tasks, the angular range of motion (ROM), displacement of the pelvis, and the thorax rigid bodies, as well as ROM of spine angles between the initial and final position of the subject, were investigated in the plane in which motion was performed (frontal for lateral bending, sagittal for forward bending and transverse for rotations). Antero-Posterior (AP) and Medio-Lateral (ML) CoP displacements during the task were also investigated. The mean of each dependent variable was considered. All variables were expressed as mean ± standard error (SE) after verification of normal distribution and equal variance with the Shapiro–Wilk test. To analyze presumed differences for the static and forward bending tasks, 3 groups (HS, AIS-LL, AIS-RT) ANOVAs were applied to the dependent variables. For lateral bending and rotation movement tasks, 3 groups (HS, AIS-LL, AIS-RT) × 2 sides (left, right) ANOVAs were performed.

## 3. Results

The results are summarized in Table 1.

There was no significant difference between groups in terms of age, height, and weight. AIS-LL and AIS-RT groups exhibited similar Cobb angles and skeletal maturity based on the Risser sign. 

Regarding the static task, the analysis of the 95% confidence ellipse area showed a main effect on the group. Sway area was significantly larger for AIS-RT than for the HS group, the AIS-LL group exhibiting an intermediary behavior. The mean speed of CoP displacements and sample entropy showed no significant effect between groups. Analysis of the static segmental position of the upper body with respect to the reference frame showed a main effect of group for pelvis axial rotation (transverse angle). The amplitude of pelvis rotation was larger for AIS-RT than for the AIS-LL group. HS group was not different from both AIS groups. A second main effect of the group was present for sagittal kyphosis angle. It was significantly lower for the AIS-RT group than for both HS and AIS-LL groups. Finally, although no significant effect of group for bi-acromial line rotation relatively to the pelvis was observed, around one-third of all AIS patients showed a rotation of the bi-acromial line relatively to the pelvis greater than 5 deg. When present, this trunk rotation was always toward the left. 

In lateral bending, analysis of kinematic data (Figure 2) showed a significant main effect of group for pelvis ROM. It was lower for both AIS groups than for HS group. In addition, the ML amplitude of CoP displacements also showed a significant effect on the group with an ML amplitude higher for the HS than for both AIS groups. A significant effect of group was also present for frontal spine angle, which was lower for the AIS-RT group than for HS and AIS-LL groups. Finally, pelvis and thorax frontal displacement both revealed a main effect of the group. Post-hoc tests showed that contralateral pelvis displacement was significantly larger for AIS-LL than for HS, and that thorax displacement was significantly lower for AIS-RT than for the HS group. Despite small differences between the AIS-LL and AIS-RT groups, reported differences in lateral bending reflected a different bending strategy present in both AIS groups in comparison to the HS group. While HS rotated their spine mainly around the pelvis, AIS subjects performed the bending so that the fixed rotation point during bending was localized approximately at L2 (Figure 2). Neither side nor interaction effects were significant.

No significant differences were observed in rotation. However, ROM and displacements were consistently greater for HS than for AIS patients. In forward bending tasks, no differences were noted.

LA = left acromion, RA = right acromion, T1 = first thoracic spinous process, KA = kyphosis apex, TLJ = thoracolumbar junction, LoA = lordosis apex, S = sacrum, LPSIS = left posterior-superior iliac spine, RPSIS = right posterior-superior iliac spine, RASIS = right antero-superior iliac spine, LASIS = left antero-superior iliac spine, XP = xyphoïd process, JN = jugular notch, SKA = sagittal kyphosis angle, SIA = sagittal inflexion angle, SLA = sagittal lordosis angle, FKA = frontal kyphosis angle, FIA = frontal inflexion angle, FLA = frontal lordosis angle, and FAA = frontal biacromial angle.

## 4. Discussion

The main objective of this study was to compare trunk motion and posture between mild AIS and healthy children when performing simple uniplanar upper body movements. We showed that AIS-RT patients exhibited a greater sway area than HS, confirming previous results [4,5], while AIS-LL exhibited an intermediary behavior. CoP speed and sample entropy did not show any difference, suggesting that for the three groups postural control required a similar amount of muscular activity and attention. Regarding upper body kinematics during the static task, we showed that mild scoliosis only sparsely affects body symmetry. However, when an asymmetry was observed, it was constantly in the same direction, with a trunk torsion toward the left with respect to the pelvis as already reported in previous studies for severe AIS-RT patients [4,5] and related to the Cobb angle [10]. In our cohort, this trunk torsion, when present, was independent of AIS type and not linked to the Cobb angle and could be considered a predictive factor of scoliosis aggravation rather than severity. This result supports the findings from Pesenti et al. [17] who recorded similar torsion behavior during gait regardless of the curve type. The results also highlighted a reduced kyphosis in AIS-RT with respect to healthy participants, that is, a decreased physiological sagittal curvature near the main scoliotic curvature—well known as the flat-back syndrome.

Although several studies have described the alteration in gait patterns in scoliosis patients when compared to normal individuals [18], there is a lack of data when focusing on static tasks. Our results suggest that AIS patients, whatever the type of scoliosis, showed a reduced pelvic rotation suggesting a reduced range of trunk motion in the frontal plane as reported during gait [19]. It could be due to an increased body stiffness because of 3D structural changes in the pelvis [20] and/or considered as a compensation for postural imbalance and/or preserve patients from pain issues. Interestingly, lateral bending differences in pelvis and thorax displacements were indicative of different biomechanical strategies that seem similar in AIS-LL and AIS-RT. While bending, AIS performed a contralateral displacement of the pelvis and smaller displacement of the thorax so that the fixed rotation point during bending was localized approximately at L2, whereas HS rotated mainly around the pelvis (Figure 2). This strategy may be used to compensate for the postural imbalance. Scoliotic patients would try limiting CoP displacements to avoid a potentially critical position as confirmed by the higher lateral amplitude of CoP displacements for healthy subjects. However, this strategy was not observable for frontal bending, certainly, because it is a symmetrical movement less destabilizing. Muscle tone asymmetry of the erector spinae may also play a role during side bending [21]. 

Contrary to our hypotheses, no asymmetry in right and left motions were observed in either AIS-LL or AIS-RT group, suggesting a similar and symmetric kinematic behavior whatever the side of the curvature. The curve size of our patients may be also not sufficient to observe sub-group differences between curve types. Nevertheless, this indicated that the observed kinematic alterations are rather independent of static deformation and already present at an early stage of AIS suggesting that they could be investigated as markers for early detection of AIS and/or aggravation. 

The main limitation of this study is the small sample size of each group. Although the observed differences are statistically significant, our findings require confirmation with a wider cohort of patients. Only mild scoliosis was included in the study. Different types of curve magnitude will need to be analyzed in the future to investigate whether trunk motion contains progression predictive parameters. However, the kinematic analysis seems to be an efficient tool to characterize AIS. Further investigations using multivariate methods on larger cohorts are necessary to better characterize AIS motion and come up with a more satisfying three-dimensional classification of AIS, and potential markers for diagnosis. 

This study is a plea for a routine spine motion analysis. Motion capture is a noninvasive way to assess the impact of spine deformities closer to daily life than static standing X-rays. Combining different modalities of spine analyses will greatly improve our knowledge of spine deformities and AIS, particularly for detection, progression predictive factors, and therapy planification. 

## Figures and Tables

**Figure 1 children-09-00738-f001:**
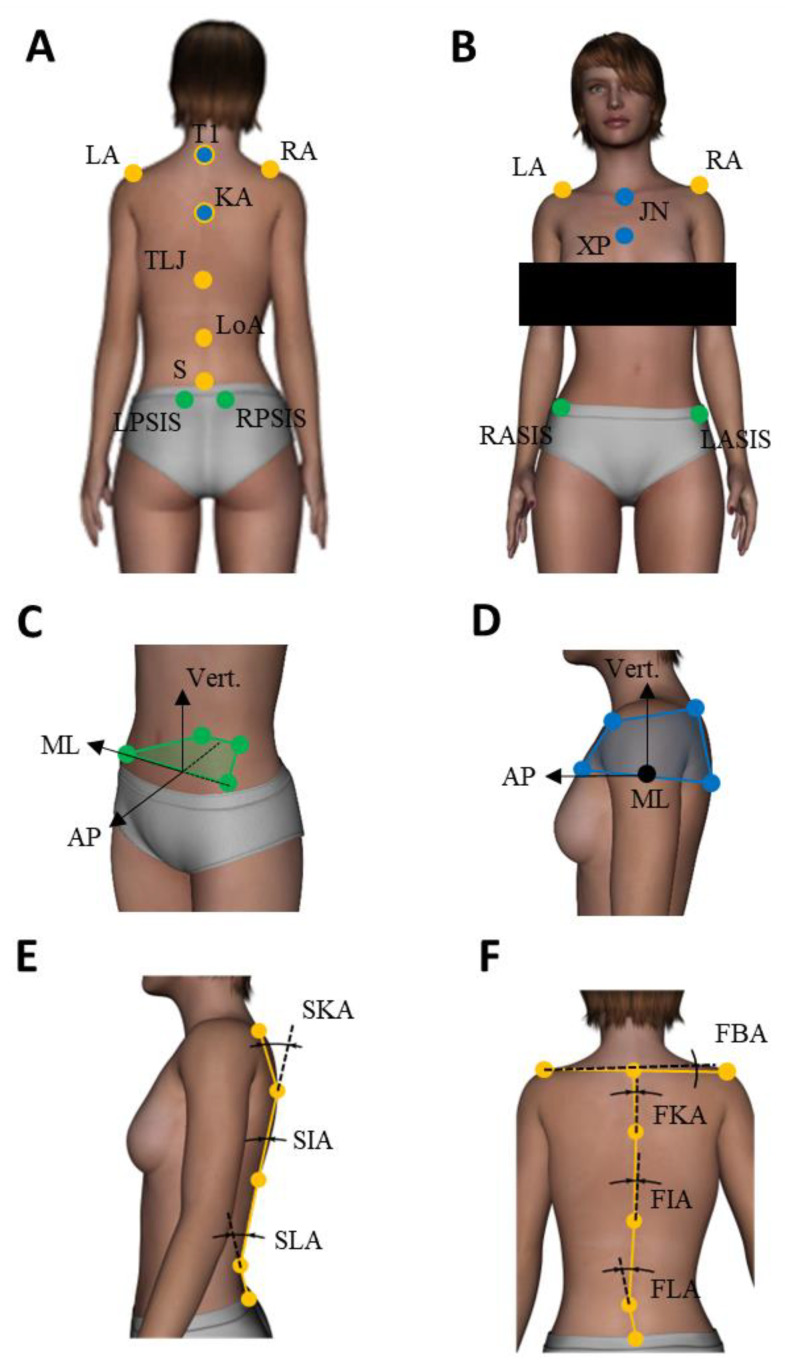
Reflective marker positions and dependent variables definition. Back (**A**) and front (**B**) views of markers location. The reference frame was defined by its antero-posterior axis (AP), the horizontal symmetry axis between both feet, its medio-lateral axis (ML) perpendicular to the AP axis in the horizontal plane, and its vertical axis, normal to the horizontal plane pointing upwards. (**C**) Definition of a pelvic rigid body (green markers): ML axis is defined as the axis connecting RASIS and LASIS, the vertical axis is normal to the plane containing RASIS, LASIS, and the middle of RPSIS and LPSIS then the AP axis is the vector product between ML and vertical axis. (**D**) Definition of a thoracic rigid body (blue markers): vertical axis is defined as the axis connecting the middle of XP and KP and the middle of T1 and JN, the ML axis is normal to the plane containing T1, JN, and the middle between XP and KM, then AP axis is the vector product between ML and the vertical axis. (**E**) Definition of the sagittal angles of the spine (yellow markers). (**F**) Definition of the frontal angles of the spine and of the biacromial line, connecting LA and RA (yellow markers). Note that the transverse rotation of the biacromial line and the global angle of the spine with respect to the vertical in sagittal and transverse planes, which have also been investigated are not represented here.

**Figure 2 children-09-00738-f002:**
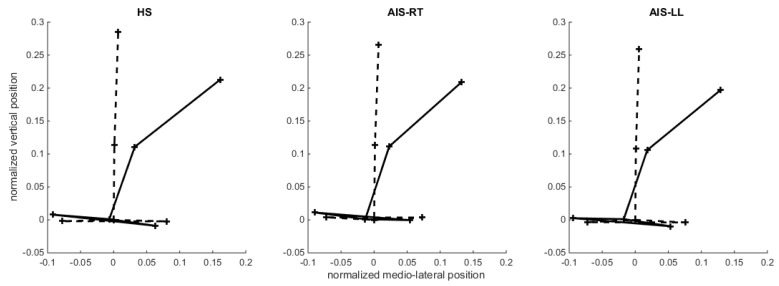
Back view of the spine in lateral bending. Plot of the back view (frontal) of the mean normalized positions of the spine (sacrum, inflexion point, and T1) and the pelvis (antero-superior iliac spines) at the beginning (dotted line) and at the end (solid line) of the lateral bending for the HS, AIS-LL, and AIS-RT groups.

**Table 1 children-09-00738-t001:** Summary of results. In lateral bending and rotation tasks, values from right and left movements were averages since no effect of side was observed. F: Frontal, S: Sagittal, T: Transverse.

	Main Effect (Group)		Parameters Values (Mean ± Std Err)						HSD Post-hoc *p* Values		
	F	*p*	HS (n = 15)		LL (n = 15)		RT (n = 17)		HS × LL	HS × RT	LL × RT
Anthropometric data											
Age (years)	0.16	0.85	13.0	±1.9	12.3	±1.8	12.6	±1.1			
Weight (kg)	2.36	0.11	49.0	±9.4	40.9	±8.5	47.1	±6.8			
Height (cm)	0.58	0.57	156.4	±9.3	152.4	±7.2	158.0	±8.2			
Cobb Angle (°)	1.04	0.31			20.1	±4.1	20.8	±4.2			
Risser sign	0.44	0.51			1.9	±1.7	1.9	±1.6			
*Static task*											
Ellipse (mm²)	3.70	0.03	155.8	±14.8	194.5	±22.2	232.7	±28.8	0.49	0.03	0.48
Mean Speed of CoP	2.08	0.14	27.1	±1.1	30.0	±1.4	27.8	±0.5			
Sample entropy	1.16	0.32	0.58	±0.1	0.50	±0.1	0.40	±0.1			
Pelvis F angle (°)	0.62	0.54	−0.2	±0.6	−1.2	±0.5	−0.9	±0.8			
Pelvis S angle (°)	1.73	0.19	−2.4	±1.7	−3.0	±2.3	1.2	±1.3			
Pelvis T angle (°)	3.60	0.04	0.1	±0.9	0.0	±0.9	−2.7	±1.0	0.99	0.08	0.04
Thorax F angle (°)	0.27	0.76	−0.5	±0.7	0.0	±0.9	−0.9	±1.1			
Thorax S angle (°)	0.10	0.91	1.9	±2.7	2.5	±2.8	1.0	±1.8			
Thorax T angle (°)	3.21	0.05	−0.4	±0.7	−0.1	±0.9	2.7	±1.1			
Acr. line F angle (°)	0.05	0.95	1.9	±0.4	2.8	±0.5	2.6	±0.4			
Acr. line T angle (°)	1.70	0.19	1.9	±0.3	3.6	±0.7	4.1	±0.6			
Spine F Angle (°)	0.28	0.76	1.0	±0.4	1.2	±0.5	0.7	±0.6			
Spine S Angle (°)	0.77	0.47	5.5	±0.6	4.4	±0.6	4.7	±0.6			
Lordosis F angle (°)	0.29	0.75	−2.1	±2.1	−2.1	±2.3	−4.2	±2.3			
Lordosis S angle (°)	1.94	0.16	33.8	±2.0	28.4	±2.4	31.7	±1.3			
Inflexion F angle (°)	0.40	0.67	0.1	±1.1	1.3	±1.2	1.6	±1.4			
Inflexion S angle (°)	0.96	0.39	−6.0	±1.5	−4.6	±1.8	−2.8	±1.6			
Kyphosis F angle (°)	1.02	0.37	0.7	±1.2	−0.8	±1.6	2.0	±1.4			
Kyphosis S angle (°)	6.13	0.00	21.2	±1.5	20.9	±1.3	14.9	±1.6	0.99	0.01	0.01
*Lateral bending*											
ML CoP disp.	5.68	0.00	59.8	±3.5	47.3	±3.1	50.9	±4.0	0.00	0.05	0.58
Pelvis F ROM (°)	4.67	0.01	6.1	±0.5	4.7	±0.5	4.7	±0.5	0.03	0.02	0.99
Thorax F ROM (°)	0.63	0.53	45.2	±1.5	44.1	±1.8	43.3	±1.8			
Pelvis F disp. (×10^−3^)	4.19	0.02	12.7	±2.5	20.7	±3.2	16.0	±2.4	0.01	0.42	0.19
Thorax F disp. (×10^−3^)	4.47	0.01	165.4	±4.9	158.4	±6.4	148.4	±5.9	0.48	0.01	0.19
Spine F angle (°)	8.43	0.00	37.3	±1.2	36.5	±0.8	33.1	±1.3	0.79	0.00	0.01
Lordosis F angle (°)	1.74	0.18	20.9	±1.2	19.9	±1.9	18.4	±1.2			
Inflexion F angle (°)	0.54	0.58	15.8	±1.1	16.4	±1.1	16.9	±1.1			
Kyphosis F angle (°)	1.89	0.16	13.1	±1.6	10.8	±2.2	9.8	±1.4			
*Rotations*											
CoP displacement	2.81	0.07	29.4	±2.5	24.0	±2.3	26.6	±1.8			
Pelvis T ROM (°)	1.39	0.25	20.1	±3.4	15.8	±3.4	15.4	±2.5			
Thorax T ROM (°)	1.08	0.34	40.6	±1.8	40.2	±2.6	37.7	±2.1			
Pelvis T disp. (×10^−3^)	1.47	0.24	32.0	±4.4	28.3	±3.3	25.9	±2.9			
Thorax T disp. (×10^−3^)	1.97	0.15	49.2	±5.6	43.3	±5.0	40.6	±3.0			
*Frontal bending*											
AP CoP displacement	0.67	0.52	64.2	±5.5	63.9	±6.8	72.8	±6.1			
Pelvis S ROM (°)	0.35	0.71	36.1	±3.4	39.0	±3.1	39.5	±2.8			
Pelvis S disp. (×10^−3^)	0.08	0.93	46.3	±4.2	44.0	±4.1	45.5	±4.5			
Thorax S disp.(×10^−3^)	0.38	0.68	420.8	±8.9	414.3	±11.3	426.6	±9.2			
Spine S Angle (°)	0.12	0.89	93.5	±3.3	95.3	±3.0	95.3	±2.6			
Lordosis S angle (°)	1.32	0.28	46.0	±1.7	41.9	±2.1	45.7	±1.9			
Inflexion S angle (°)	0.54	0.59	18.4	±1.2	16.8	±1.9	16.5	±1.1			
Kyphosis S angle (°)	2.03	0.14	2.5	±1.4	6.2	±1.4	3.5	±1.2			

## Data Availability

Not applicable.

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
