# Peer review of "Comparison of Trunk Motion between Moderate AIS and Healthy Children"

_children, 2022, doi:10.3390/children9050738_

Round 1

Reviewer 1 Report

The article is quite interesting. However, it needs to be improved.
Please clearly state the objectives of the research and the hypotheses.
Please expand the Discussion, see the article: Wilczyński J. Electromyographic activity of the erector spinae and convexity as well as concavity of spinal curvature in children. Children 2021, 8 (12), 1168; https://doi.org/10.3390/children8121168.
The presentation of the results is correct.
The application should be linked with the title of the article and the purpose of the research.

Author Response

The article is quite interesting. However, it needs to be improved.

Please clearly state the objectives of the research and the hypotheses.

The objectives and the hypothesis were improved in the intro.

Please expand the Discussion, see the article: Wilczyński J. Electromyographic activity of the erector spinae and convexity as well as concavity of spinal curvature in children. Children 2021, 8 (12), 1168; https://doi.org/10.3390/children8121168.

This reference was added in the text

The presentation of the results is correct.

Thanks

The application should be linked with the title of the article and the purpose of the research.

The title and the purpose of the research were improved

Reviewer 2 Report

I would like to thank the editor for the opportunity to review this manuscript. This article aimed at investigating kinematic differences and asymmetries in early AIS in a static task and in uniplanar trunk movements. The aim of the study is interesting but, prior of the publication, I suggest several revisions.

first of all, I suggest modifying the title with more information about the type of the study.

Introduction

I suggest to expand the scientific rationale.

Materials and Methods

I suggest specifying where participants were enrolled and where the study took place

I suggest explaining the characteristics of the room where the study was carried out

Line 55, I suggest to insert the adherence to the Helsinki Declaration (add reference)

Line 61 and 65, I suggest clarifying how the data of the 4 tests were processed to obtain the value used for the statistical analysis

Line 84, I suggest to insert the full definition before the ISBN acronym

Line 93, I suggest to insert the full definition before AP and ML acronyms

I suggest moving lines from 102 to 104 to line 53, before EC approval

Discussion

I suggest increasing the comparison between the results and previous studies

I suggest entering the study limits, if any

I suggest to insert why the results of this study are important and the applicability of findings

Author Response

first of all, I suggest modifying the title with more information about the type of the study.

The title was modified

Introduction

I suggest to expand the scientific rationale.

This has been improved 

Materials and Methods

I suggest specifying where participants were enrolled and where the study took place

This was added in the text.

I suggest explaining the characteristics of the room where the study was carried out

This was added in the text.

Line 55, I suggest to insert the adherence to the Helsinki Declaration (add reference)

This was added in the text.

Line 61 and 65, I suggest clarifying how the data of the 4 tests were processed to obtain the value used for the statistical analysis

The 4 tests are described in figure 1. The reference of the figure was added in the text.

Line 84, I suggest to insert the full definition before the ISBN acronym

This was added in the text

Line 93, I suggest to insert the full definition before AP and ML acronyms

This was added in the text

I suggest moving lines from 102 to 104 to line 53, before EC approval

I am sorry, I do not understand the comment. I feel difficult for the reader to understand the statistical analysis performed on the data issued from the MoCap before we introduce the tasks.

Discussion

I suggest increasing the comparison between the results and previous studies

The discussion was improved on this point

I suggest entering the study limits, if any

Limitations were added

I suggest to insert why the results of this study are important and the applicability of findings

Perspectives were added

Round 2

Reviewer 1 Report

I accept the corrected article. It has been carefully revised and is now, in my opinion, fit for publication.